# Narratives of most significant change to explore experiences of caregivers in a caregiver-young adolescent sexual and reproductive health communication intervention in rural south-western Uganda

Cecilia Akatukwasa[1,2]*, Elizabeth Kemigisha[2,3], Dorcus Achen[2,4], Danielle Fernandes[1,4], Shakira Namatovu[2], Wendo Mlahagwa[2], Gad Ndaruhutse Ruzaaza[5], Gily Coene[2], Godfrey Zari Rukundo[6], Kristien Michielsen[1], Viola N. Nyakato[2,7]

1 Faculty of Medicine and Health Sciences, Department of Public Health and Primary Care, International Center for Reproductive Health, Ghent University, Ghent, Belgium, 2 Faculty of Interdisciplinary Studies, Mbarara University of Science & Technology, Mbarara, Uganda, 3 African Population and Health Research Center-Nairobi, Nairobi, Kenya, 4 Centre of Expertise on Gender, Diversity and Intersectionality, Vrije Universiteit Brussel, Brussels, Belgium, 5 Faculty of Medicine, Department of Community Health, Mbarara University of Science & Technology, Mbarara, Uganda, 6 Faculty of Medicine, Department of Psychiatry, Mbarara University of Science & Technology, Mbarara, Uganda, 7 The Nordic Africa Institute-Uppsala, Uppsala, Sweden

* cecilia.akatukwasa@ugent.be

## Abstract

### Background

This paper presents findings from a qualitative effectiveness evaluation of an intervention aimed at improving caregiver-young adolescent sexual and reproductive health (SRH) communication including training modules for caregivers on parent-child SRH communication.

### Methods

Data was collected (October 2021-November 2021) using a narrative interviewing technique with thirty caregivers (8 males and 22 females), who received the parent-child communication intervention in Mbarara district, south-western Uganda. We explored caregivers' experiences with the intervention based on four domains of change: caregiver-young adolescent communication on SRH issues, knowledge and attitudes towards adolescent SRH, parenting skills, and personal life and family. Thematic analysis was used to code and analyse the data, with attention to gender differences.

### Results

Findings highlight positive parenting as a key attribute of SRH communication, along with a transformation of knowledge and attitudes towards the SRH of young adolescents leading

**Data Availability Statement:** All relevant data are within the paper and its Supporting Information files.

**Funding:** Research reported in this article was supported by VLIR-UOS under the TEAM and South Initiatives projects (VLIR-UOS Team Project 2019; UG2019TEA493A103). The content of this article is solely the responsibility of the authors and does not necessarily represent the official views of VLIR-UOS. The funders had no role in study design, data collection and analysis, decision to publish, or preparation of the manuscript.

**Competing interests:** The authors have declared that no competing interests exis.t

to an overall improvement in SRH communication. However, communication is still limited to comfortable topics.

## Conclusion

Our findings indicate improved caregiver–adolescent SRH communication practices following a community intervention. Programming for adolescent health on broader sexuality topics, comfortability and attitude change among caregivers could promote behaviour change on a long term. Future studies may focus on the long term impacts of interventions of this nature and test interventions aimed at addressing comfortability with discussingSRH issues.

## Background

Sexual and Reproductive Health (SRH) education for young adolescents (10–14 years) is essential for mitigating negative SRH outcomes during adolescence [1]. Currently, HIV/AIDS, other sexually transmitted infections (STIs) and teenage pregnancy are on the rise among young adolescents in sub-Saharan Africa (SSA) [2]. School-based sexuality education strategies have been the main avenue wherein adolescents receive comprehensive SRH education. However, in many countries, this is assumed to be foreign and in conflict with acceptable traditional family norms, and has thus had a negative reception from cultural and religious institutions as well as from teachers tasked with the role of delivering comprehensive sexuality education (CSE) [3]. Leveraging the role of caregivers as the primary providers of SRH information for young adolescents is an alternative public health strategy [4]. Caregiver-adolescent communication around SRH is precisely important for realizing adolescent rights to access SRH information [5]. It equips them with the knowledge, skills, and attitude changes that enable them to make responsible SRH choices, including delay of sexual debut and choosing protected sex [5,6].

However, parental SRH communication in sub-Saharan Africa is marred by intergenerational conflict. This is a potential source of tension between the caregiver and young adolescent [7]. The cultural and religious contexts of families and communities further hinder SRH communication [8–10]. In Uganda, communicating sexuality issues to adolescents was formerly the role of paternal aunts and uncles. However, the change of family structure from an extended to a nuclear family has shifted the role of the primary providers of SRH information away from extended family to primary caregivers (i.e. mothers and fathers). Resource-limited settings present an additional unique context in which caregiver-child communication occurs [8]. Adolescents in these settings are disproportionately at a higher risk of SRH risks including unintended pregnancies, exposure to sexually transmitted infections (STIs) and vulnerability to sexual violence [11].

The dynamics of caregiver-adolescent communication comprise an interplay of influences such as the knowledge, attitudes, comfort level, values and beliefs of caregivers. This interplay in turn influences not only how they convey SRH information but also how adolescents receive it [12–17]. At an interpersonal level, relationship quality and parenting styles between the caregiver and the adolescent influence communication [5,18–20]. Parenting styles, specifically authoritarian ones, tend to position young adolescents as passive recipients of SRH information [21], while a warm and loving relationship between the adolescent and caregiver is a stronger foundation for good communication [22]. Gender dynamics also influence SRH communication [8,16,17]. Same sex communication is more common compared to

communication with the opposite sex [16,17]. Communication between mothers and daughters is more common compared to fathers and their sons or fathers with daughters [16].

Current evidence acknowledges the need to engage caregivers through providing both informational support and strategies for communication. These could help improve caregiver competence, efficacy, and comfort in deciphering SRH information to the young adolescents [14]. Such support and strategies should be family centric, responsive to cultural and community values, and should take into account a breadth of information as effective ways to improve communication [4,14,23]. However, current evidence indicates that most parent-child communication interventions are devoid of context-specific relevance and fail to establish how these strategies materialize within broader aspects of SRH communication and parenting styles.

Interventions to improve parent and child communication on SRH in settings similar to Uganda have shown positive results in improving SRH outcomes, especially when initiated during early adolescence [4,15]. We developed a parent-only multi-session intervention targeting caregivers of young adolescents with the primary aim of providing caregivers with knowledge about the SRH of young adolescents, imparting behavioural skills, instilling attitude change, and fostering good communication skills and positive parent-child relationships. This paper presents findings on the post-intervention evaluation examining the effectiveness and outcomes of the intervention using the most significant change (MSC) stories. These findings indicate broader experiences and changes that occurred due to the intervention beyond the specific communication on SRH (which will be presented in a separate quantitative evaluation paper).

## Methods

The study ran from October 2021 to November 2021 among caregivers selected from 6 villages in Rwebishekye parish, in Mbarara district-south-western Uganda. Among an estimated population of 6,061 people, the study community comprised approximately 1,520 households, of which 29% were headed by women [24].

### Description of the intervention

The intervention was embedded within a larger project conducted in south-western Uganda to improve caregiver and young adolescent communication on SRH. The baseline status of caregiver-young adolescent SRH communication is provided in separate publications [8,25]. Using Appreciative Inquiry (AI), a participatory research approach, we jointly developed culturally sensitive parent-child communication guidelines over which the parents would have ownership [26]. This framework allowed us to shift the current paradigm away from solely preventing negative behaviours among adolescents and towards empowering caregivers to enhance their developmental assets and abilities to make positive, informed decisions. The AI methods worked in four stages; 1) Discovery: Identify what works well through community based participatory research (CBPR) activities and baseline data collection in the formative phase; 2) Dream: Envision changes that would work well in the future in collaboration with the community stakeholders and a community advisory board (CAB) whose role was to routinely provide feedback on the most impactful practices of parent and YA communication. Initially, the CAB collaborated with the research team to interpret the barriers and facilitators to parent-young adolescent communication. 3) Design: Develop the intervention in collaboration with the stakeholders who would review the modules and provide input and feedback on its pertinence towards the study community to ensure age, cultural and religious appropriateness. 4) Destiny: Implement the proposed intervention in collaboration with the community

members and evaluate the effectiveness of the intervention. The intervention had 3 components namely: 1) the content; 2) training of community facilitators; 3) training of caregivers.

We therefore collectively developed training modules designed for caregivers of young adolescents, which community leaders twinned with research team members then delivered over the course of 15 weeks in small group sessions of 10–20 participants distributed across the six villages in the community (Table 1). The training for caregivers was delivered in two phases in April/May and September/October 2021. For each training phase, groups of parents received training two days a week (based on their preference) and each session lasted 1–2 hours each. Pre and post assessment on knowledge and attitude changes were done at the beginning and end of each training phase. Participatory learning approaches including discussions, storytelling, group work and role plays were used. The training sessions were designed to build the parenting skills of caregivers, enable parents to address structural barriers to SRH communication, improve the communication skills of parents around SRH, improve parental knowledge of young adolescent SRH, and address attitudes and normative beliefs concerning young adolescent SRH, all in order to improve caregiver comfort with SRH discussions.

## Evaluation of the intervention

We evaluated the intervention using the most significant change (MSC) technique, a form of participatory monitoring and evaluation involving the collection and selection of stories of change [27]. MSC specifically helps participants describe the changes and identify expected and unexpected changes brought by the intervention. The typical MSC process includes 1) defining the domains of change; 2) deciding on how and when to collect the change stories; 3) collecting the significant change stories; 4) selecting the most significant change stories; and 5) verifying the stories [27]. The process can be adapted depending on the nature and context of the intervention [28]. While a typical MSC activity would involve the selection of stories

**Table 1. Outline of the caregiver-young adolescent communication intervention.**

| Phase | Sessions |
|---|---|
| **Phase1-Preparation of community training** | Selection of community trainers (1 session) |
| | Facilitation skills for trainers (1 session) |
| | Training of trainers (14 sessions) |
| **Phase 2-Parenting adolescents** | Who is a parent-better parenting (3 sessions) |
| | Social, cultural and religious norms regarding parenting (2 sessions) |
| | Adolescence, why act now? (2 sessions) |
| **Phase 3-Parent and adolescent communication on SRH** | Communication skills on adolescent health (2 sessions) |
| | Puberty (2 sessions) |
| | Relationships (1 session) |
| | HIV/AIDs and other STIs (1 session) |
| | Pregnancy prevention (1 session) |
| **Phase 4-Community talks on safety and prevention of selected adolescent health risks** | Sexuality and sexual behaviour (1 session) |
| | Sexual violence, recognition and reporting (1 session) |
| | Gender, sexuality and human rights (1 session) |
| | Emotional and mental health–building self-esteem and addressing depression and addiction (1 session |
| | Entrepreneurship skills for parents (1 session) |

through a hierarchical process within different levels of authority [27,29], we chose to analyse all the stories that were collected in the study given that each of them presents a unique experience. Prior to the collection of the significant change stories, four domains of change were pre-defined, based on the project's objectives: a) caregiver and young adolescent communication on SRH issues; b) caregiver knowledge and attitudes towards adolescent SRH; c) parenting skills; and d) personal life and family. However, we also considered domains of change that emerged during analysis of the stories.

## Selection of study participants and story collection

We purposively selected caregivers of young adolescents from the communities where the intervention was implemented, identified by the community leaders in the respective villages. Three trained researchers collected stories in the local dialect of the area (Runyankore-Rukiga), one month after concluding the implementation of the intervention. We used a narrative approach to explore the experiences of study participants based on the four domains of change (a–d, above). The researchers were trained in qualitative research methods including the most significant change technique. After consenting participants, the researchers collected the stories at a convenient location in the community. Each story collection session lasted between 10 and 30 minutes. The stories were audio-recorded to capture participants' narratives verbatim. However, at the end of each interview, the researchers probed and reflected back on the story to seek clarity on certain aspects as reported by the participants. The audio files were transcribed verbatim in the local language and then translated into English. We then convened a plenary focus group of nine participants to discuss and verify whether the stories accurately reflected changes in the community.

## Data analysis

We adopted a thematic analysis approach to interpret the change stories. We adopted a hybrid of deductive and inductive analysis to generate themes from the data [28,30]. Based on the six stages of thematic analysis suggested by Braun and Clarke, the researchers iteratively read the change stories to gain a deeper understanding of the data until we reached saturation. [30]. This allowed us to examine the different types of changes beyond the set domains of change and thus gain a full picture of the impact of the intervention [28].

We applied an initial coding framework to the significant change stories to study fragments of data including words, lines and segments to understand their meaning [30]. We followed this with focused coding, where the initial codes were clustered into the *a priori* domains of change already mentioned, and we noted emergent themes. Below, we present the results of both emergent and *a priori* thematic areas [30].

## Ethical consideration

Ethical clearance was obtained from the Research Ethics' committee of Mbarara university of Science and Technology (15/05-19) and the Uganda National Council of Science and Technology (UNCST) (SS 5108). All participants provided written informed consent to participate in the study during enrolment. Permission was also obtained to allow the story to be recorded. Privacy and confidentiality were maintained during the data collection process. This was achieved by selecting a convenient location for the interview within the community where felt comfortable to express their views. Each participant was assigned a unique identification number and no identifiers were associated with the participants' stories.

## Results

Thirty caregivers were enrolled onto the study. These comprised 8 (26.7%) male caregivers and 22 (73.3%) female caregivers. Their median age was 45 years. Table 2 is a summary of the results presented in line with the domains of change.

### Improved caregiver-young adolescent SRH communication

**Overall improvement in communicating SRH topics.** Parents reported communicating about SRH topics like general health and body hygiene, menstruation and menstruation hygiene, wet dreams, body changes and the implications of these changes. In some instances, caregivers communicated to their children about the availability of STI screening services as well as contraception for adolescents. They also reported having discussions around pregnancy and other negative consequences of engaging in risky sexual practices.

Some narratives indicated that pubertal changes were a trigger for communicating about SRH risks like unwanted pregnancy or HIV. Caregivers reported engaging their children in discussions by encouraging them to approach them in case of any concerns, especially those related to menarche. A female caregiver shared her story of change, emphasizing not just the topics discussed with her children, but also how she felt relaxed as she communicated.

"*My children had not started engaging in sex, but I took time and talked to them about STDs and HIV. I asked them if they knew about AIDS and if they know that it kills. One of my children said she had cut herself and asked if that meant she was going to suffer from HIV/AIDS. I told her that it is possible to get AIDS through cuts, but the most important thing right now is to protect themselves especially now that they have already started menstruating because they can become pregnant and also suffer from HIV/AIDS. Once, we were weeding the garden, I told them that 'do you know that this person almost died*?' *They asked me the cause which I said was AIDS. I also told them that condoms can prevent one from getting pregnant. I talk to them freely because we are now used to each other. They ask me questions based on what they know from school, which was not the case in the past because I used to fear telling them about such. The more they ask me the more I use the chance to explain to them and get a lot to talk about. They can only ask you questions when they don't fear you.*" (Female caregiver)

However, SRH communication was still marred by threats as when caregivers reported the use of scare tactics to pass on SRH information. Discussions with boys revolved around warnings against engaging in sexual relationships or else facing the consequences of unwanted pregnancies like being incarcerated.

**Table 2. Summary of results.**

| Themes /domains of change | Sub-themes |
|---|---|
| Improved caregiver and young adolescent communication | • Improvement in the combination of SRH topics• Comfort with SRH communication• Gender matching in SRH communication |
| Adopting positive parent and child relationships to improve SRH communication | • Positive parenting skills• Encouraging adolescent's agency and negotiating good behavior• Transformation of gendered perspectives in parent-adolescent relations |
| Personal and family life | • Improved couple relations• Adoption of entrepreneurship skills |
| Community support and community level change | • Collective child rearing practices<br>• Improved community connectivity and creation of community support systems for parenting |

**Comfort with SRH communication.** Caregivers reported being more comfortable and less anxious or embarrassed discussing SRH with their young adolescents after the intervention. More so, their children were less fearful when approaching them about SRH and sensitive issues regarding their sexuality. Having developed warmer and friendlier relationships with the young adolescents after the intervention, adolescents could approach caregivers even about the risk of sexual violence in the community, in addition to experiences of bodily changes like menarche. Adolescents could ably seek material support during menstruation like pads and knickers which had not been the case before the intervention. Caregivers recalled their prior experiences of SRH communication with their older adolescents as being uncomfortable. A parent narrated being previously embarrassed to discuss some sensitive topics such as family planning and condoms, but with the intervention, they were more comfortable. A female caregiver shares her experience prior and after the intervention;

"*At the start of the training, I would feel shy and wonder how I would mention some difficult terms in front of my children. But now, all is well because we are very close to each other. This is because of the strategies we were taught to use to approach our children. Now they tell me all their problems without fear and I also guide them accordingly.*" *(Female caregiver)*

**Gender-sensitive SRH communication.** Caregivers highlighted challenging cultural norms around gender roles and cross-gender communication. Some caregivers acknowledged they could communicate to their child on SRH regardless of gender differences. This is mainly reported by the male participants. A male caregiver shared his experience of talking to his daughter about menstruation in the absence her mother and supporting her with her menstruation needs. Another acknowledged that despite being a man, the new communication skills acquired from the intervention enabled him to talk to his daughter without having to go through her mother first.

"*I had learned and obtained the experience on how to communicate with my daughter without any fear of being a man. This had actually always been hard for me to do before I took part in the study. This has helped very much in preventing her from the bad conduct that was developing.*" *(Male caregiver 1)*

"*Before, I never cared so much about my child, or even never cared that she had become an adolescent. I didn't even understand much about adolescence. But ever since the intervention started, I learned that there's a lot a male parent contributes to the growth of a child, and when I understood this, I started paying attention to my daughters, especially this one who is ten years. . .Recently she got into her first period and her mother wasn't around but she came running and told me "while I was urinating, I urinated blood". I quickly went and bought pads for her. This all happened because I had talked to her about her menstruation just like we were taught during the intervention.*" *(Male caregiver 2)*

A female caregiver also appreciates that she has learnt that it is also important to talk to her son about SRH specifically their pubertal changes as well as the fact that they are also at risk of SRH problems if they engage in risky sexual behaviour.

**Transformation in knowledge and attitudes towards SRH.** Parents acknowledged their role as the primary SRH communicators with their young adolescents; post-intervention. They appreciated the caregivers' significant role in helping young adolescents navigate pubertal changes. Prior to the study, they believed that sexuality education was solely provided at

school, while concurrently expressing their uncertainties on the accuracy of the SRH information provided there. Caregivers now acknowledge their cardinal role in deciphering SRH information for their young adolescent children. Additionally, they appreciate the fact that they can competently talk about SRH with young adolescents of the opposite sex without having to go through their partners. Fathers appreciate being able to talk to their daughters while mothers appreciate being able to talk to their sons about their SRH matters.

The stories also indicated increased knowledge among caregivers around different aspects of young adolescent SRH. One caregiver disclosed that before the intervention, he did not pay attention to his children as they entered adolescence, holding onto the belief that they were still too young to receive SRH information. After the training, they reported being more attentive to their children's pubertal changes. One caregiver appreciated learning from the intervention that their young adolescent is at risk of getting pregnant since she previously believed that only older adolescents were capable of getting pregnant. Learning this prompted her to talk to her child about bodily changes such as menarche at age 12 years and the possibility of getting pregnant even then.

> "*As a parent I never knew that a girl of 12 years can go into her menstrual periods or become pregnant. I always looked at her as a child but after attending the training, we now sit together and I tell them everything, even if one is [only] 13 years old. I tell her that "boys can impregnate you." I go ahead to tell her about her body's changes.*" (Female caregiver)

Caregivers also expressed a more positive attitude towards communicating sensitive SRH issues such as condom use. Several caregivers also mentioned learning more about how to handle instances of sexual violence. One caregiver revealed learning about the process to undertake in case of such violence, such as the need for the adolescent to be examined by a health worker and the provision of preventive therapy for HIV. They also learned that young adolescents can be screened for STIs and that adolescents also need information on these services.

> "*My most interesting topic and what I learned the most is a case where a young girl who stays with the grandmother or mother or relative is sent for something but is raped along the way. We always knew that the first thing is to run to the LC1 (Local Council) and police and report the case. But we instead found out that rushing this child to the hospital would save her from many things like pregnancy [and] HIV/AIDS. I loved this so much, because we didn't know about it as village people.*" (Female caregiver aged 45 years)

### Adopting positive parent and child relationships to improve SRH communication

**Positive parenting skills.** Caregivers indicated an improved responsiveness to their children by acknowledging that their role goes beyond giving birth to the child to include nurturing them and having a positive and friendly relationship with them. Caregivers report more warmth and support towards their children after the intervention. Caregivers transformed their relationships with their children from authoritarian and neglectful parenting to more affective and hands-on parenting styles. Caregivers adopted a friendlier and composed approach to relating with their children, who in turn reciprocated that warmth of feeling. This is manifested through reports of prioritizing spending more time with their children, treating them with affection and communicating with them in a calmer manner. As mentioned above, they also began to pay greater attention to the physical needs of their children, such as sanitary

pads and underwear for girls, to prevent them from obtaining these through other transactional relationships that may put them at risk.

Adolescents began to openly talk about their personal lives with their caregivers, which was not the case prior to the intervention. Due to the friendly atmosphere, adolescents no longer sought information from third parties in the community but readily approached their caregivers without fear. More so, they admitted that this friendly atmosphere had helped to erase their fear of talking about SRH with their children. Some caregivers report adopting some techniques for creating comfortable and friendly environments to be able to have conversations with their children like taking advantage of meal times in the living rooms. These improvements also carried over into relationships between caregivers and their non-biological children.

**Encouraging children's agency and negotiating good behaviour.** Caregivers described how by changing their attitudes to create a less fearful environment and allowing open and comfortable discussions on different issues including sexual and reproductive health, they can now have friendly conversations with their children, listening to their opinions and allowing them the opportunity to make decisions at the household level. Young adolescents began to participate independently in domestic activities, taking on roles and executing them with minimum supervision from the caregivers.

They report abandoning punitive approaches including corporal punishment and arguing with their children, some of which were adopted from how they themselves were parented, to take up negotiating good behaviour through open and positive communication while employing listening skills. They also reported experiences of providing social rewards to young adolescents who exhibit good behaviour. This has enabled them to monitor their children, especially their movements, establish where they spend their time, who they spend their time with, how they spend their day in a fearless environment. This has earned them a positive change in their children's behaviour allowing them to be more open and transparent about what they do in the absence of their parent.

"*I always make some popcorns at home and tell them 'whoever comes back home early will eat the popcorn and those that will come late will find that the popcorn is finished.' This makes them come back home early. Also, because I am close with my children, they now tell me the challenges they face. I also tell them that "you see you have matured and your breasts have developed, so if you allow a boy to touch your breasts or sleep with him, you will become pregnant, get HIV and die.*" (Female caregiver)

**Transformation of gendered parental roles in parent-child relationships.** Caregivers reveal a change in attitude towards parenting roles. They now understand the notion of equal parenting roles no matter the issues and the sex of the child. This is mainly reported among male caregivers. A male caregiver reports previous experience of using third party communication, i.e. Communication to his children through his wife. After the intervention, he can now directly talk to his children, especially his daughter. Additionally, they have reformed traditional gender attitudes to begin socialising male young adolescents to take on household roles traditionally meant for females.

"*My boy is 12 years old and I see that he has changed positively because he now knows that boys can also peel plantains [plantain is a traditional food in Uganda-before cooking, it must be peeled and traditionally, it is role of the girl to complete this task–peeling (plantain) is not for just girls*! *A boy can also mop the house.*" (Female caregiver)

**Personal and family life.**   Participants report a warmer relationship with their partners and overall improved harmony in families. Couples can communicate with each other on different matters especially those regarding their children and the household. They report being able to sit and plan together which was not the case previously, when individuals abandoned their parenting roles to their partners creating tensions within households. The training on entrepreneurship and financial literacy empowered parents with new skills on how to improve their household income and support their children.

"*Before I used to say, 'I earned the money myself and I can use it the way I want since I am the family head, without consulting my wife.' If I got like UGX150,000, I used to buy 3kgs of meat for my family to also enjoy, not knowing that my wife maybe could have a loan somewhere and could be paid using this same money and the meat waits until next Saturday. But after the study sessions, I now engage my wife in planning for the money that I get so that in case she has something bigger than what I had to spend this money on, then we go for that first, and this study has helped me so much with doing this.*"(Male caregiver)

**Community support and community level changes.**   Adoption of collective community child-rearing practices was a key change at the community level. Parents reported that they had spread the knowledge they acquired to other children in the community whose caregivers were not part of the intervention. Caregivers also reported making a resolution to improve community connectivity and create community support systems for parenting. Having conducted the intervention amidst the Covid-19 restrictions, participants report that the community did not have incidents of unwanted pregnancies in contrast to many other communities. This was attributed to the intervention.

## Discussion

This qualitative evaluation study assessed the effectiveness of a community-based intervention aimed at improving caregiver-young adolescent communication on SRH. The most significant change stories highlight an improvement in caregiver-young adolescent SRH communication as a result of the intervention. Specific improvements included; caregivers broadening the range of topics discussed with young adolescents, an improvement in caregiver knowledge of young adolescent SRH, and increased comfort discussing SRH with young adolescents. Improvement in parent-child relationships and the adoption of positive parenting practices was also a very salient finding from the significant change stories.

Overall, the intervention addressed key contextual and underlying issues affecting SRH communication beyond the generic and superficial parameters constantly cited as key influencers of SRH communication such as socioeconomic status, gender, level of knowledge among others. This intervention is unique in a setting where the sexuality of young people is a sensitive matter due to religious and cultural dispositions, making open discussions about sex taboo [12]. Findings prior to the development of this intervention demonstrated that the level of SRH communication increases with greater comfort around SRH discussions [25]. This intervention addressed underlying issues affecting SRH communication including parenting styles and strained parent-child relationships that mediate between comfort around having SRH discussions and the actual experiences of SRH communication. Many participants reported feeling more comfortable and open in their discussions about SRH after participating in the intervention. Some caregivers reported feeling more at ease discussing sensitive topics such as contraception.

In this study, SRH encompassed ten broad SRH topics: general health and body hygiene; menstruation and menstruation hygiene; nocturnal emissions in boys; HIV/AIDS and other STIs; handling sexual pressure; sexual conduct; having babies and birth control; romantic relationships; condoms; and sexual violence and reporting. Prior evidence indicated that belief in the importance of SRH communication was an important mediator for parent-young adolescent SRH communication [31]. However, despite reports of increased SRH communication, the stories of change indicated that discussions remained narrowly focused on avoiding potential negative consequences of sex, such as HIV/AIDS, other STIs, and unwanted pregnancies. The conversations are still marred by warnings, threats and spread of fear as is illustrated in previous findings [10]. Indeed, the conversations seemed to avoid explicit discussions of ways to mitigate or prevent these risks. The persistent lack of comprehensiveness in SRH discussions can be explained by the societies' cultural and religious norms in the study setting that prohibit open discussions surrounding sex [10]. On the other hand, general health and body hygiene as well as menstruation and menstruation hygiene are prominently discussed. Overall, caregivers not only acknowledged the benefit of discussing SRH with young adolescents but importantly expressed willingness to have these discussions after the intervention despite the dearth in the breadth of the topics.

The stories of change indicated an increased knowledge of the SRH of young adolescents. Knowledge of SRH is an important factor as far as SRH communication is concerned. Caregivers who had a high SRH knowledge were more likely to have adolescents who adopted positive SRH behaviours [32]. Lack of knowledge prior to the intervention was tied to beliefs about the timing of SRH communication, and the onset of puberty; many caregivers felt that the age category (10–14 years) was too young to initiate conversations around sex [33]. Participants reported gaining a better understanding of SRH topics and feeling more informed about the risks and benefits of sexual behaviour.

Gender norms have an influence on the content of SRH discussions. The stories of change reflected a transformation of gender norms and roles during SRH communication. Prior findings on gender and its effects on SRH communication revealed that SRH communication was most common between mothers and their daughters, rather than between mothers and their sons or fathers and their sons [8,16]. Previous studies have also revealed the traditional role of mothers as the primary socializing agents as far as SRH is concerned [8]. In this intervention, stereotypes around gender roles were left behind, especially among the men who initially believed that discussions around SRH topics such as menstruation were the role of the female caregiver. Male caregivers revealed affirmative experiences in adopting these roles of talking to their daughters.

One of the most notable effects of the intervention is the great improvement in parent-child relationships. Caregivers acknowledged their initial deficiencies in parenting as well as the strained relationships they had with their children, and how this intervention successfully addressed these challenges. Changes in parenting styles and the enhancement of parent-child relationships have been indicative of promoting good and open SRH communication [22]. Positive parent-child relationships and closeness fosters open relationships, and thus open and comfortable discussions between caregivers and young adolescents about SRH [10,34]. Further, a warm relationship between the caregiver and young adolescents, builds agency and promotes decision-making skills in sexual relationships. Studies show that agency, or more specifically, motivational autonomy, is a critical resource in attaining positive reproductive outcomes for adolescents [35]. Not coincidentally, this agency also facilitates adolescents' rejection of traditional gender norms [35], norms which caregivers also attempted to challenge as a result of the intervention.

Community-wide interventions on caregiver-child communication around SRH have shown promising results among caregivers of adolescents [4]. It was therefore imperative to utilize the MSC technique; a participatory approach to evaluate this intervention. The MSC technique as an evaluation approach in this study augments the quantitative findings of the end-line evaluation for this project. The MSC technique specifically helps to identify underlying as well as unexpected changes brought about by the intervention beyond the set domains of change [27]. Previous literature indicates that MSC may not be used as a stand-alone method of evaluation given that it may not capture certain types of changes or may help to capture changes that would rather not be captured using a different approach [28].

Our evaluation has some limitations. Stories from 30 participants may not necessarily present the views of all the other community members that participated in the intervention. More so, the participants were selected from a single community, which may limit the generalizability of the findings. This being a participatory research project required a high level of rapport building. Given its duration in the community, coupled with frequent collaboration with community members, that rapport building may have created significant social desirability biases in our participants' responses. Participants were overwhelmingly appreciative of the project with mostly positive stories of change. They were less critical of the intervention. Additionally, the stories report the immediate and short-term effects of the intervention rather than its long-term effects, which remain to be studied. The MSC technique in this study diverts from the original design especially in the selection process of the stories. Nevertheless, the narrative nature of the evaluation allows us to capture first-hand, relatively open and spontaneous expressions of our participants' experiences of intervention-inspired changes, in addition to those underlying changes that may not be captured using other structured evaluation approaches.

## Conclusion

This study assessed the effectiveness of a community-based intervention aimed at improving caregiver-young adolescent communication. Our findings suggest that a community-based multi-session training program for caregivers of young adolescents produced positive effects in SRH communication. Caregivers adopted skills not only directly relating to SRH communication but also skills known to mediate SRH communication such as parenting styles, knowledge and attitudes towards SRH, and comfort around having SRH discussions. Future evaluations should explore the long-term impacts of interventions of this nature. Implementation studies should look for ways to scale-up community-based interventions in the wider population to test their application in different settings especially as these concern policies and guidelines on parenting in resource-limited, small communities, with cultural and religious taboos around discussing sex. More specifically, such research can focus on empowering parents to promote decision-making skills and motivational autonomy among their adolescent children. Additional future interventions can focus on best practices to normalize open discussions around SRH and dispel deeply ingrained beliefs around sexuality that have been shown to deter positive SRH outcomes for adolescents.

## Supporting information

**S1 Checklist.**
(TXT)

**S1 File.**
(DOCX)

## Acknowledgments

The authors wish to acknowledge the contribution of the Community Advisory Board and community stakeholders for their invaluable feedback on the research process. We are also grateful for the contribution of the research team, the community mobilizers and all research participants. The authors also wish to thank Jason Johnson Peretz for clarifying the language of the manuscript.

## Author Contributions

**Conceptualization:** Elizabeth Kemigisha, Wendo Mlahagwa, Gily Coene, Godfrey Zari Rukundo, Kristien Michielsen, Viola N. Nyakato.

**Data curation:** Cecilia Akatukwasa, Shakira Namatovu.

**Formal analysis:** Cecilia Akatukwasa, Dorcus Achen, Viola N. Nyakato.

**Funding acquisition:** Gily Coene, Kristien Michielsen, Viola N. Nyakato.

**Investigation:** Cecilia Akatukwasa, Elizabeth Kemigisha, Wendo Mlahagwa, Gad Ndaruhutse Ruzaaza, Gily Coene, Godfrey Zari Rukundo, Kristien Michielsen, Viola N. Nyakato.

**Methodology:** Cecilia Akatukwasa, Elizabeth Kemigisha, Gily Coene, Kristien Michielsen, Viola N. Nyakato.

**Project administration:** Elizabeth Kemigisha, Kristien Michielsen, Viola N. Nyakato.

**Resources:** Elizabeth Kemigisha, Viola N. Nyakato.

**Software:** Cecilia Akatukwasa.

**Supervision:** Elizabeth Kemigisha, Godfrey Zari Rukundo, Kristien Michielsen, Viola N. Nyakato.

**Validation:** Cecilia Akatukwasa, Danielle Fernandes.

**Visualization:** Viola N. Nyakato.

**Writing – original draft:** Cecilia Akatukwasa, Godfrey Zari Rukundo, Kristien Michielsen, Viola N. Nyakato.

**Writing – review & editing:** Cecilia Akatukwasa, Elizabeth Kemigisha, Dorcus Achen, Danielle Fernandes, Shakira Namatovu, Wendo Mlahagwa, Gad Ndaruhutse Ruzaaza, Gily Coene, Godfrey Zari Rukundo, Kristien Michielsen, Viola N. Nyakato.

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
