## [Decision Letter · Decision Letter 0]

8 Mar 2023

PONE-D-23-03407Narratives of most significant change to explore experiences of caregivers in a caregiver-young adolescent sexual and reproductive health communication intervention in rural southwestern UgandaPLOS ONE

Dear Dr. Akatukwasa,

Thank you for submitting your manuscript to PLOS ONE. After careful consideration, we feel that it has merit but does not fully meet PLOS ONE’s publication criteria as it currently stands. Therefore, we invite you to submit a revised version of the manuscript that addresses the points raised during the review process.

We look forward to receiving your revised manuscript.

Kind regards,

Khem Narayan Pokhrel, Ph.D.

Academic Editor

PLOS ONE

Journal Requirements:

"Research reported in this article was supported by VLIR-UOS under the TEAM and South Initiatives projects (VLIR-UOS Team Project 2019; UG2019TEA493A103). VNN and KM were the primary recipients of the grant. The content of this article is solely the responsibility of the authors and does not necessarily represent the official views of VLIR-UOS. None of the sponsors played a role in the study design, data collection and analysis, interpretation of data, preparation of the manuscript, as well as the decision to submit the manuscript."

Reviewers' comments:

Reviewer's Responses to Questions

**Comments to the Author**

1. Is the manuscript technically sound, and do the data support the conclusions?

Reviewer #1: Partly

Reviewer #2: Yes

2. Has the statistical analysis been performed appropriately and rigorously? 

Reviewer #1: N/A

Reviewer #2: N/A

3. Have the authors made all data underlying the findings in their manuscript fully available?

Reviewer #1: Yes

Reviewer #2: No

4. Is the manuscript presented in an intelligible fashion and written in standard English?

Reviewer #1: No

Reviewer #2: No

5. Review Comments to the Author

Reviewer #1: Thank you for the opportunity to review this manuscript. MSC is indeed an insightful way to collect the authentic voices from the communities to understand the process of change better.

However, some comments could enhance the quality of the paper.

General comments

-It feels like the authors submitted the manuscript in hurry. There are many grammatical, punctuation, and spelling errors throughout the text. Please thoroughly proofread the manuscript before submitting the revision.

-Please use the standard format for multiple citations. Multiple citations should be inside a single parenthesis, separated by commas.

-Please use the COREQ guideline or any other standard guideline for reporting qualitative studies. Please also include the COREQ or any other applicable study checklist

-Please also highlight the areas that need improvement. Currently, it only highlights positive changes, however, some original quotes reflect areas that need improvement in communication. Please think through it, and explain these issues in the results section, and discussion section accordingly.

Specific comments

Abstract:

-The abstract should be self-explanatory on the take-home messages.

-Please mention the study site, where are these participants from?

-Also, add a sentence to elaborate on the type of parent-child communication intervention. Who received the intervention? What was the type of intervention? Was it training? Or some other behavior change communication interventions?

-Please also highlight the key findings. What kind of gaps in which domains were the most important ones?

-What were the most significant changes identified? Please be more specific with some significant examples of positive parenting and transformation of knowledge and attitudes towards SRH.

Introduction

-Please define what you mean by “young adolescents”

-Which age group are you referring to?

-While SRH is a common acronym, it is better to avoid other less common acronyms such as “YA” throughout the text

Line 56

“changes that enable them make” please add “to” - “changes that enable them to make”

-Line 57 and 61 and throughout the text– use the correct format for citation, multiple citations should be inside a single parenthesis separated by a comma

-Line 92 – the information on the age group of the young adolescent should be introduced sooner, when the term is introduced the first time, probably in line 46 in the first line

-Line 79- elaborate more on context – how does gender influence SRH communication? Is it difficult for a female to communicate more openly as compared to a male? Or it’s the same for all genders?

Methods

Description of the intervention

-Lines 119-128: for stages of AI, please avoid using the underscore sign “_”. Please replace it with “-“ or “:” or commas, and semicolons.

-Line119 – provide the full form for CBPR

-Please provide a reference for stages of AI

-Line 144- please add a comma or connecting words after MSC “The intervention was evaluated using the most significant change (MSC) technique a form of participatory”

Selection of study participants and story collection:

-The usual process of MSC involves multiple steps. Each step comprises collecting the stories and shortlisting “the most significant” stories. Right now the stories reflect the change but how are these most significant? Who was involved in deciding which stories were most significant isn’t clear. Therefore, please elaborate on the process. The selection of the most significant story should be participatory involving the storytellers also participating to decide which stories are most significant.

Results

-Please provide a table on the themes, subthemes, and categories/codes

-Please also provide a table or add a paragraph on the general characteristics of the participants- such as gender, age groups, and other relevant characteristics. The readers would want to have some basic idea about what these participants were like.

-Line 274 Please use “son” instead of “boy child”

-There are many errors in punctuation, grammar, and spelling throughout. Please proofread it thoroughly.

-The narratives can be refined and shortened without losing their authenticity

-Line 363 -365 - “you see you have matured and your breasts have developed, so if you allow a boy to touch your breasts or sleep with him, you will become pregnant, get HIV, and die.”

This statement doesn’t sound like positive parenting. While it is honest and authentic, this doesn’t reflect positive parenting. It feels like a statement instilling fear. Positive parenting should encourage safe practices and not instill fear.

Therefore, while presenting such narratives, the authors could also honestly reflect that while the communication has progressed positively, there is still room for improvement for better communication. MSC is an approach that also highlights negative changes and changes that still need improvement. Thus while describing it, please don’t just focus on the positives only. that still need improvement. Thus while describing it, please don’t just focus on the positives only.

Discussion

-Please also highlight the areas that need improvement.

-How the future interventions can be improved based on the results of this study?

Reviewer #2: The authors need to revise their manuscript. Furthermore, they need to attach the MSC interview guide and informed consent form. There are some rooms of improvement in terms of grammar, spelling, sentence structure and and punctuation. Please go through the reviewer's report for detail comments and feedback.

6. PLOS authors have the option to publish the peer review history of their article (what does this mean?). If published, this will include your full peer review and any attached files.

Reviewer #1: No

Reviewer #2: **Yes: **GEHA NATH KHANAL

---

## [Author Response · Author response to Decision Letter 0]

28 Apr 2023

24th April, 2023

Dear Editor and editorial board of Plos One

Thank you so much for the review of our manuscript “Narratives of most significant change to explore experiences of caregivers in a caregiver-young adolescent sexual and reproductive health communication intervention in rural south-western Uganda.” We also appreciate the opportunity to respond to the comments raised by the reviewers. We have revised our manuscript as per the reviewers’ comments and provided a point by point response as indicated below. The changes are also highlighted in the manuscript through track changes.

We appreciate your continued consideration of our manuscript and look forward to getting your views with regards to our revision. Am happy to respond to any questions regarding our revision and edits. 

Best wishes 

Cecilia Akatukwasa 

Corresponding author

Editor’s comments 

The style requirements have been attended to. 

This has been indicated

This will be rectified during submission

"Research reported in this article was supported by VLIR-UOS under the TEAM and South Initiatives projects (VLIR-UOS Team Project 2019; UG2019TEA493A103). VNN and KM were the primary recipients of the grant. The content of this article is solely the responsibility of the authors and does not necessarily represent the official views of VLIR-UOS. None of the sponsors played a role in the study design, data collection and analysis, interpretation of data, preparation of the manuscript, as well as the decision to submit the manuscript."

This has been revised accordingly

A codebook comprising all the significant change stories organized according to the domains of change has been included. 

The Ethics statement has been included in the methods’ section only and deleted from the other parts of the manuscript

Reviewer comments 

General comments 

Reviewer 1: 

Comment Response 

It feels like the authors submitted the manuscript in hurry. There are many grammatical, punctuation, and spelling errors throughout the text. Please thoroughly proofread the manuscript before submitting the revision.

 Thank you for this feedback. The manuscript has been thoroughly proofread to correct the grammatical errors. The article has been further sent to a third party with scientific writing expertise to conduct further language edits. 

Please use the standard format for multiple citations. Multiple citations should be inside a single parenthesis, separated by commas.

 Thank you for noting this. The citations have been rectified to include a single parenthesis separated by commas. 

Please use the COREQ guideline or any other standard guideline for reporting qualitative studies. 

 We have ensured that the COREQ guidelines are followed in the manuscript and a checklist has been attached.

Please also include the COREQ or any other applicable study checklist

 The checklist has been included as a supplementary file

Please also highlight the areas that need improvement. Currently, it only highlights positive changes, however, some original quotes reflect areas that need improvement in communication. Please think through it, and explain these issues in the results section, and discussion section accordingly.

 The areas that need improvement have been indicated in the discussion section as well as the conclusion. Line 444-459 talk about some of the negative aspects of the intervention outcomes and recommendations are provided in the conclusion section 

Specific comments

Abstract

The abstract should be self-explanatory on the take-home messages.

 I have added a sentence to the conclusion to augment the conclusion and take home message (line 44-48)

Please mention the study site, where are these participants from?

 I have mentioned the study site in line 33. 

Also, add a sentence to elaborate on the type of parent-child communication intervention. Who received the intervention? What was the type of intervention? Was it training? Or some other behavior change communication interventions?

 The sentence indicating who received the intervention and what intervention has been included line 32 and 33

Who received the intervention?

 We have included the caregivers in line 32 as the recipients of the intervention 

What was the type of intervention?

 This has been included in line 33. The intervention is a training on parent adolescent SRH communication using community participatory approaches. We used training modules that were jointly developed and adapted by community facilitators and university researchers with feedback on relevance and contextualization by a community advisory board that has representation of different members of society

Please also highlight the key findings. What kind of gaps in which domains were the most important ones?

 In line 40, the most salient domain is mentioned 

What were the most significant changes identified? Please be more specific with some significant examples of positive parenting and transformation of knowledge and attitudes towards SRH.

 These are indicated in line 40-42

Introduction

Please define what you mean by “young adolescents”

 This has been included ‘Young adolescents (10-14 years)-line 52. 

Which age group are you referring to?

 Young adolescents are those aged 10-14 years-line 52

While SRH is a common acronym, it is better to avoid other less common acronyms such as “YA” throughout the text

 Effort has been made to reduce the usage of the acronym YA throughout the manuscript

Line 56: “changes that enable them make” please add “to” - “changes that enable them to make”

 This has been rectified in line 64

Line 57 and 61 and throughout the text– use the correct format for citation, multiple citations should be inside a single parenthesis separated by a comma

 This has been rectified in the entire document 

Line 92 – the information on the age group of the young adolescent should be introduced sooner, when the term is introduced the first time, probably in line 46 in the first line

 This has been adjusted accordingly in line 103. Information on the age-group is provided in line 52

Line 79- elaborate more on context – how does gender influence SRH communication? Is it difficult for a female to communicate more openly as compared to a male? Or it’s the same for all genders?

 This has been explained in line 86-89, highlighting that communication between same sex is more common compared to communication with the opposite sex. And that communication with mothers and daughters was more common compared to fathers and sons as well as fathers and daughters. This goes with supporting literature.

Methods

Description of the intervention

Lines 119-128: for stages of AI, please avoid using the underscore sign “_”. Please replace it with “- “or “:” or commas, and semicolons.

This has been adjusted accordingly on line 117-138

Line119 – provide the full form for CBPR

 The full form of CBPR has been provided in line 126-127 as community based participatory research 

Please provide a reference for stages of AI

 This has been provided in line 122 (Article 26 by Stowell Frank) 

Line 144- please add a comma or connecting words after MSC “The intervention was evaluated using the most significant change (MSC) technique a form of participatory”

 The comma has been added in line 156

Results

Please provide a table on the themes, subthemes, and categories/codes The table has been added and in marked as table 2 in line 214

Please also provide a table or add a paragraph on the general characteristics of the participants- such as gender, age groups, and other relevant characteristics. The readers would want to have some basic idea about what these participants were like.

 The paragraph has been added to explain the participant characteristics as an introduction of the results’ section line 212-213.

Line 274 Please use “son” instead of “boy child” This has been rectified in line 291

There are many errors in punctuation, grammar, and spelling throughout. Please proofread it thoroughly. The article has been proof read for grammar, punctuation and spelling with the help of an independent reviewer.

The narratives can be refined and shortened without losing their authenticity

 We have made effort to shorten some of the narratives. However, most narratives can only speak to the theme in their detailed form and so it might be difficult to cut some of them down. 

Line 363 -365 - “you see you have matured and your breasts have developed, so if you allow a boy to touch your breasts or sleep with him, you will become pregnant, get HIV, and die.” This statement doesn’t sound like positive parenting. While it is honest and authentic, this doesn’t reflect positive parenting. It feels like a statement instilling fear. Positive parenting should encourage safe practices and not instill fear.

 It is true the narrative indicates fear, however the narrative is reporting that parents can communicate. I have included in the discussion, that although communication is taking place, it is still marred by fear and scare tactics. (Line 451-453).

Therefore, while presenting such narratives, the authors could also honestly reflect that while the communication has progressed positively, there is still room for improvement for better communication. MSC is an approach that also highlights negative changes and changes that still need improvement. Thus while describing it, please don’t just focus on the positives only. That still need improvement. Thus while describing it, please don’t just focus on the positives only.

 In line 508 onwards of the discussion, we detail the limitations of our findings. We present the fact that the participatory nature of the study and overstay as well as over engagement with the community may have built rapport. Therefore, we most definitely experienced a lot of social desirability bias and therefore could hardly document any negative outcomes of the intervention. 

Discussion 

Please also highlight the areas that need improvement These have been highlighted in the conclusion (Line 522-537)

How the future interventions can be improved based on the results of this study? This has been indicated in the entire discussion (Line 522-537)

Reviewer #2: The authors need to revise their manuscript. Furthermore, they need to attach the MSC interview guide and informed consent form. There are some rooms of improvement in terms of grammar, spelling, sentence structure and punctuation. Please go through the reviewer's report for detail comments and feedback.

Reviewer 2 Report: 

I am writing this report as a reviewer of the manuscript entitled “Narratives of most significant change to explore experiences of caregivers in a caregiver-young adolescent sexual and reproductive health communication intervention in rural southwestern Uganda”. 

Using most significant change stories, the authors have tried to explore the experience of SRH communication intervention among the caregivers of young adolescents. The study found several interesting findings like improvements in communication between the caregivers and young adolescents, improvement in positive parenting skills, increased harmony between the caregivers and YAs, changing attitude towards gender roles among the caregivers who attended the training interventions. 

I would suggest couple of recommendations/suggestion for further improvement of this paper. However, authors can use their discretion to agree or disagree with the feedback and consider the revision.

General Comments 

Comment Response 

YA, YA children, children and adolescents are being used interchangeably throughout the manuscript. I would suggest making consistency. 

 Thank you for noting this. This has been rectified in the entire document, whenever relevant

I was interested to know what the interventions were. Although table 1 presents the outline of the intervention, I was interested to know more about the intervention. 

 The details of the interventions are provided in a detailed training manual that would be available upon request. It indicates the objectives of each session, the content of the sessions and the methods of delivery. It may not be possible to detail these in the manuscript. 

Training modules were delivered through small group sessions (10-20 participants) for 15 weeks, I was interested to know how many days in a week? How many hours in each session? What was the teaching learning approach? How was the review done? Thus, I would suggest presenting more detail of intervention somewhere in the supplementary files so that it would be beneficial for those who are implementing similar programs.

 The training for parents was delivered in two phases in April/May and September/October 2021. For each training phase, groups of parents received training on two days a week (based on their preference) and each session lasted 1-2 hours each. Pre and post assessment on knowledge and attitude changes were done at the beginning and end of each training phase. Participatory learning approaches including discussions, story telling, group work and role plays were used. 

1. The authors need to justify the number of stories they collected, and they included for the analysis. There are some possible biases that might have occurred while conducting this study. How have the authors addressed the bias?

I. Possible bias after selecting the participant purposively and selecting all the stories that were collected, 

In line 162-166, we mention that the MSC process was adapted. We did not follow the typical MSC process and this is also indicated in the study limitations in the discussion section

II. Although the authors have mentioned about their limitation in the discussion section, I was interested how the participants/respondents were identified by the researcher who was an external 

 The participants were not selected by the researcher but rather community leaders (line 185-187

Selecting the researchers for collecting the stories (Possible bias towards the views/text from the researchers who are good at story telling.

 Line 186-189, we indicate the verification of the stories by members of the community to reduce this bias.

Make consistency in using active/passive voices throughout the paragraph, (Example Line 181-183 or tense (Example; Line 187-188), line 253 or punctuation or spelling (example line 233, 153).

 The document has been revised and the following major revisions have been made:

1. Changing passive voice to active voice.

2. Eliminating ‘there is’ and ‘there are’.

3. Condensing phrasal descriptions to adjectival ones (e.g. SRH of young adolescents � young adolescent SRH)

The manuscript needs paraphrasing and making simple sentences in several places to make the readers understand easily. 

 This has been done.

Check the referencing and citation style for this Journal and revise it accordingly 

 The journal recommends the use of Vancouver style of referencing. This has been adapted in the manuscript and thus the reference list updated. 

Specific Comments 

Section Line # Comments/Feedback Response

Abstract There is unclarity between background, methods and findings. The authors need to clarify this

 The different sections of the abstract have been clarified. The introduction, methods, results and conclusion are presented in distinct sections. 

Background 46 What do the authors mean to mention by YA here? It would be better if there is definition of YA in the introductory paragraph. Although the authors have defined YA as children aged 10-14 years in line # 92, it would better to be mentioned here in the introductory paragraph 

 This has been included in the first paragraph (Line 52).

 49-52 Can make two sentences for more clarity to the readers.

 The sentences have been split 

 52, 69 Full form of CSE, Use abbreviation in line #69 

 This has been rectified (line 60)

 74 What do you mean by quality of relationship here?

 This implies how good or bad the relationship between the caregiver and the child is.

 What is the rationale for using MSC as a technique for this study? Line 159-160: indicates the rationale for using the MSC technique

Methods 105 Mean age of caregivers should be in findings rather than the method An introduction section to the results has been introduced to describe the characteristics of the study participants including the median age line 213-214.

 107 Is it the population estimate for 2022 or the findings of census data of 2019? It is 2016-that is the most recent available document from Uganda Bureau of statistics 

 114 How the guidelines were culturally sensitive? What culturally sensitive issues do those guidelines mean to address? The context of the study is clearly stated in the introduction section. Here we mention that open discussions about sex are culturally tagged as taboo, thus making this a sensitive topic. The guidelines elucidate this by adopting participatory methodologies which are stated in the method’s section. 

 144 Is there any rationale for using MSC technique for evaluating the intervention? Line 149-150 gives the rationale of MSC. Line 480 to 503 details the application of MSC in this study, why it was important to use the technique despite the many limitations it presented during the study.

 173 Who were included in the plenary focus group for verifying the stories?

 The plenary focus group discussion comprised of selected community stakeholders.

 194-195 How was the privacy and confidentiality maintained during the data collection process? What was the setting of the interview? This has been added to the ethical considerations statement (line 207-209)

 Was the data analysis done manually or was done by using software? 

 The data was coded manually. However the codes/ domains of change were agreed on by the entire study team

Results Socio-demographic profile of respondents is required here. Please move the age-related information from methods to result section.

 This has been included line 213-214

 366 The authors reported changing attitude towards traditional gender roles of children (372-374). What was the intervention that the caregivers did with there children to change such gender norms? Was it observed in more than one stories? 

 Several stories of change indicate this. Issues on gender are mainly embedded in the training modules. The training manual will be attached as a supplementary file 

 I suggest the authors to present the domains of change in a figure somewhere in the result section so that the readers will easily understand 

 This has been summarized in table 2

Discussion Since, the findings are based on the predetermined domains of change (line 166), triangulating the findings with quantitative findings could have been better. Although the authors have triangulated the findings from quantitative study (Reference #26), more discussion of such findings and other quantitative programmatic data might be required

 We have results from an effectiveness study but are not published yet. The findings in references #25 are baseline findings and partly inform the basis for intervention development

References Some of the references are incorrect or incomplete (Reference #9). Please check all the references are correctly cited. The reference list has been reviewed and updated.

---

## [Editor Report · Decision Letter 1]

15 May 2023

Narratives of most significant change to explore experiences of caregivers in a caregiver-young adolescent sexual and reproductive health communication intervention in rural south-western Uganda

PONE-D-23-03407R1

Dear Dr. Akatukwasa,

We’re pleased to inform you that your manuscript has been judged scientifically suitable for publication and will be formally accepted for publication once it meets all outstanding technical requirements.

Kind regards,

Khem Narayan Pokhrel, Ph.D.

Academic Editor

PLOS ONE
---

## [Editor Report · Acceptance letter]

22 May 2023

PONE-D-23-03407R1 

Narratives of most significant change to explore experiences of caregivers in a caregiver-young adolescent sexual and reproductive health communication intervention in rural south-western Uganda 

Dear Dr. Akatukwasa:

I'm pleased to inform you that your manuscript has been deemed suitable for publication in PLOS ONE. Congratulations! Your manuscript is now with our production department. 

Kind regards, 

on behalf of

Dr. Khem Narayan Pokhrel 

Academic Editor

PLOS ONE